# Co-evolution within structured bacterial communities results in multiple expansion of CRISPR loci and enhanced immunity

Nora C Pyenson[1], Luciano A Marraffini[1,2]*

[1]Laboratory of Bacteriology, The Rockefeller University, New York, United States; [2]Howard Hughes Medical Institute, The Rockefeller University, New York, United States

**Abstract** Type II CRISPR-Cas systems provide immunity against phages and plasmids that infect bacteria through the insertion of a short sequence from the invader's genome, known as the 'spacer', into the CRISPR locus. Spacers are transcribed into guide RNAs that direct the Cas9 nuclease to its target on the invader. In liquid cultures, most bacteria acquire a single spacer. Multiple spacer integration is a rare event which significance for immunity is poorly understood. Here, we found that when phage infections occur on solid media, a high proportion of the surviving colonies display complex morphologies that contain cells with multiple spacers. This is the result of the viral-host co-evolution, in which the immunity provided by the initial acquired spacer is easily overcome by escaper phages. Our results reveal the versatility of CRISPR-Cas immunity, which can respond with both single or multiple spacer acquisition schemes to solve challenges presented by different environments.

*For correspondence: marraffini@rockefeller.edu

## Introduction

Clustered, regularly interspaced, short, palindromic repeats (CRISPR) loci and their associated (*cas*) genes provide immunity against viruses (*Barrangou et al., 2007*) and plasmids (*Marraffini and Sontheimer, 2008*) that infect prokaryotes. Phage- and plasmid-derived 'spacer' sequences that are intercalated (*Bolotin et al., 2005*; *Mojica et al., 2005*; *Pourcel et al., 2005*) between repeats provide the specificity of the CRISPR-Cas immune response. This is because spacers are transcribed and processed into small RNAs (known as CRISPR RNAs, crRNAs) that guide Cas nucleases (*Brouns et al., 2008*) to destroy complementary nucleic acids of the invaders (*Abudayyeh et al., 2016*; *Garneau et al., 2010*; *Samai et al., 2015*; *Westra et al., 2012*; *Zetsche et al., 2015*). New spacers are acquired at a low rate from the invader's genome upon infection and, remarkably, are invariably inserted in the first position of the CRISPR array (*Barrangou et al., 2007*). Therefore, spacer acquisition is comparable to an immunization of the bacterial or archaeal host, with the CRISPR array providing a temporal 'vaccination' record of the immunization events.

Depending on the *cas* gene content of a CRISPR locus, CRISPR systems are classified into six different types (I-VI) (*Koonin et al., 2017*). Here, we investigated spacer acquisition by the *Streptococcus pyogenes* and *Streptococcus thermophilus* type II CRISPR-Cas systems, which employs the RNA-guided nuclease Cas9 to provide immunity through cleavage of the invader's DNA (*Garneau et al., 2010*; *Sapranauskas et al., 2011*). This nuclease cleaves DNA targets that contain (i) a 20-nt complementary sequence with the crRNA guide, known as the protospacer, and (ii) a purine-rich sequence motif located immediately downstream of the protospacer, known as the protospacer-adjacent motif (PAM) (*Gasiunas et al., 2012*; *Jinek et al., 2012*). While these targeting rules are required to ensure that the response is specific against the virus, they also represent a liability to the host since phages

carrying mutations in either the 'seed' sequence of the protospacer (a critical region of 6–8 nt at the 3' end) or PAM can escape the type II CRISPR defenses (*Deveau et al., 2008*).

Our lab studies spacer acquisition in *Staphylococcus aureus* liquid cultures expressing the type II-A CRISPR-Cas locus of *Streptococcus pyogenes* SF370, after infection with the φNM4γ4 phage (*Heler et al., 2015*). In this system, approximately 1 in $10^7$ non-immune infected cells survive through the incorporation of a new spacer, which occurs immediately upon infection and preferentially from the injected double-stranded DNA end of the phage (*Modell et al., 2017*). The acquisition rate can be substantially increased in immune cells, where Cas9 cleavage of the viral DNA generates additional double-stranded DNA ends for the spacer acquisition machinery, a phenomenon known as 'priming' (*Nussenzweig et al., 2019*). The viral population often contains escaper phages that overcome the immunity mediated by a given guide RNA and can lyse bacteria harboring the cognate spacer. However, escaper phage propagation comes to a halt when they infect a host that contains a different spacer sequence (*van Houte et al., 2016*). Most staphylococci are immunized only once (*Heler et al., 2015*; *Heler et al., 2019*), therefore the spacer diversity required for the neutralization of escaper phage is achieved at the population, as opposed to the cellular, level. A small fraction of staphylococci that acquire multiple spacers can also be found, however whether this feature is important for survival is poorly understood.

We wondered how the growth on solid media would affect spacer acquisition. In this environment, bacteria that acquire new spacers grow into separated colonies that cannot mix to neutralize escapers. Therefore, it would be reasonable to expect that the rise of escapers within colonies would quickly end with the annihilation of the host. Indeed, theoretical studies have shown that due to the physical separation and clustering of different type of resistant hosts, a structured environment reduces the effective diversity of host resistance and promotes pathogen emergence (*Chabas et al., 2018*). However, many CRISPR-resistant colonies can be formed after infection of staphylococci or streptococci with φNM4γ4 or φ2972 phage, respectively (*Barrangou et al., 2007*; *Heler et al., 2015*; *Heler et al., 2017*), results that suggest that the type II CRISPR-Cas immune response can limit the emergence of CRISPR-escaping viruses when cells are growing in solid media.

Here, we studied the formation of phage-resistant colonies in top agar media, using both our heterologous type II system and *S. thermophilus* cells that naturally encode this system. We found that the fate of the colony depends on the strength of the first spacer acquired. If the founder spacer has a very low proportion of escapers in the phage population, the colony grows smoothly and maintains a simple structure with the majority of cells harboring only this spacer. In contrast, when the viral population contains a high frequency of target mutations that avoid the type II CRISPR-Cas immunity mediated by the founder spacer, sectored colonies are formed, with each sector being composed of cells harboring additional spacers acquired after the founder spacer. In these colonies, the new spacers are selected by the escaper phage, which lyses bacteria containing only the founder spacer and thus promotes the formation of sectored colonies. While the colonies originated by weak founder spacers struggle to rise when compared to those founded by strong spacers, they undergo a co-evolution with the phage that leads to the generation of spacer diversity and higher viral resistance. Our study reveals the versatility of type II CRISPR-Cas systems, which can respond with both single or multiple spacer acquisition modes to generate immunological diversity at the population or cellular level, respectively, and ensure survival in different environments.

## Results

### Bacteriophage-resistant colonies display two modes of spacer acquisition

After infection of a liquid culture of *Staphylococcus aureus* RN4220 cells (*Kreiswirth et al., 1983*) carrying the type II-A CRISPR-Cas locus of *Streptococcus pyogenes* SF370 (*Deltcheva et al., 2011*) in the plasmid pC194 (*Horinouchi and Weisblum, 1982*) with the φNM4γ4 phage (*Goldberg et al., 2014*), bacterial survival is achieved through the acquisition of spacer sequences from the viral genome (*Heler et al., 2015*), with the new spacer inserted at the leader-end of the CRISPR array (*Figure 1—figure supplement 1A*). Amplification of the CRISPR locus from the entire culture shows two PCR bands, one corresponding to bacteria that did not acquire new spacers and most likely succumbed to phage infection, and another corresponding to the acquisition of a single spacer

(*Figure 1A*). When we plated the infected culture to analyze the CRISPR locus of survivors, we confirmed that most cells acquired a single spacer, with occasional cases of two and three spacer acquisition events (*Figure 1B*). Previous studies that performed next-generation sequencing of the expanded CRISPR array showed that thousands of different spacer sequences are acquired in this experimental system (*Heler et al., 2015*; *Heler et al., 2019*). This high diversity of CRISPR targets is required to neutralize the rise of 'escaper' phages; that is phages containing mutations that avoid targeting by a given spacer sequence (*van Houte et al., 2016*).

A key aspect of this mechanism of neutralization is the continuous mixing of the liquid culture. This distributes cells containing different spacers and ensures that escaper viruses are unable to spread. We wondered whether and how escapers are neutralized in a more structured environment where phage and CRISPR-adapted cells cannot mix. To test this, we performed the same experiment described above, but in semi-solid agar plates, mixing bacteria and phage at a multiplicity of infection, MOI, of 2: $2 \times 10^9$ staphylococci and $4 \times 10^9$ $\phi$NM4$\gamma$4 particles, conditions that minimize the rise of non-CRISPR resistance. About 2000 colonies grew after 48 hr of incubation, which we checked

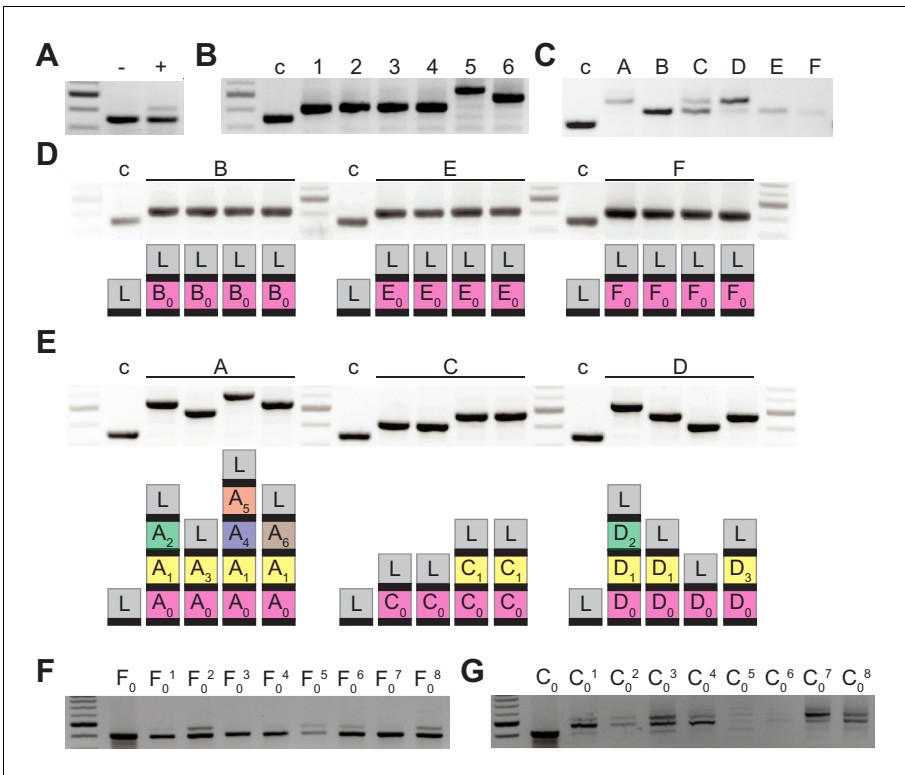

**Figure 1.** Bacteriophage-resistant colonies display two modes of spacer acquisition, determined by the sequence of the founder spacer. Agarose gel electrophoresis of the products of the amplification of the pCRISPR array present in staphylococci infected with $\phi$NM4$\gamma$4, using plasmid DNA templates extracted from: (**A**) liquid cultures with (+) or without (-) addition of phage; (**B**) six colonies formed after plating the infected liquid cultures, 'c' indicates a no-spacer control sample; (**C**) bacteriophage-resistant colonies (labeled with different upper case letters) after infection in semi-solid (top agar) media, 'c' indicates a no-spacer control sample; (**D**) colonies resulting from the re-streak of mono-spacer colonies shown in (**C**), with a schematic of the pCRISPR array determined after sequencing of the PCR product (L, leader; black rectangle, repeat; colored boxes, acquired spacers with different colors indicating different spacer sequences within a colony); (**E**) same as (**D**) but after re-streak of multi-spacer colonies; (**F**) eight colonies resulting from infection in semi-solid media of cells containing pCRISPR with the founder spacer F ($F_0$, pre-infection control); (**G**) same as (**F**) but for founder spacer C.
The online version of this article includes the following source data and figure supplement(s) for figure 1:

**Source data 1.** Spacer sequences from the PCR products shown in *Figure 1*, *Figure 1—figure supplement 1BC*, *Figure 2B*, *Figure 3B*, *Figure 4B–C*, *Figure 5A–B*, *Figure 5—figure supplement 1D–G*, *Figure 7A* and *Figure 7CD*.
**Figure supplement 1.** PCR analysis of the CRISPR array of re-streaked cells.

for spacer acquisition via PCR to find that some acquired a single spacer and others displayed a heterogenous composition (*Figure 1C*); that is they contained cells with 1, 2, 3 and 4 spacers. To analyze the spacer content of different cells within these colonies, we re-streaked three of them and amplified and sequenced the CRISPR locus of four of the resulting colonies. We found that the original mono-spacer colonies were composed of staphylococci harboring the same spacer sequence (*Figure 1D* and *Figure 1—source data 1*). In contrast, the multi-spacer colonies contained cells with 1, 2, 3 or 4 spacers (*Figure 1E* and *Figure 1—source data 1*). In addition, in all cases, the sequence of the first acquired spacer was the same, indicating that the founder cell of the colony acquired a single spacer that allowed phage resistance, but as this cell divided more, different, spacers were added. To corroborate that the different cells of the colony originated from the same single-spacer ancestor, we employed a version of the type II-A CRISPR plasmid containing a randomized sequence upstream of the first repeat, that functions as a unique barcode for each plasmid (*Heler et al., 2019*). We performed the same experiment using this plasmid and found a complete correlation between the barcode and first spacer sequence in each cell of the surviving colonies. This result shows (i) that the different spacers are not a consequence of the presence of different pCRISPR plasmids within the cells of the colony and (ii) that multi-spacer colonies are generated by a single cell that acquired a (founder) spacer (*Figure 1—figure supplement 1B–C* and *Figure 1—source data 1*).

## The sequence of the founder spacer determines the colony type

Next, we investigated the effect of the founder spacer on the expansion of the CRISPR array. We used plasmids isolated from one of the mono-spacer colonies or from one of the colonies that contained only the first acquired spacer obtained after re-streaking multi-spacer colonies. To eliminate any possible chromosomal mutations that could have played a role in the selection of the phage-resistant colonies, we re-introduced both plasmids into *Staphylococcus aureus* RN4220 cells and performed a modified version of the soft-agar infection experiment. These staphylococci contain a spacer that makes them immune to phage lysis, therefore, to mimic the conditions that lead to survival in the original experiment, we mixed only 2000 CRISPR-immune cells (the previously determined number of CRISPR-survivors) with $1.3 \times 10^9$ susceptible (without the CRISPR plasmid) staphylococci and $2.6 \times 10^9$ φNM4γ4 particles (MOI = 2). As expected, after 48 hr of incubation, we obtained about ~ 2000 phage-resistant colonies on each plate and analyzed the spacer content of 8 of them. We found minimal spacer acquisition in the colonies obtained from the mono-spacer founder (*Figure 1F*). In contrast, the surviving colonies produced from the multi-spacer founder plasmid displayed substantial spacer acquisition (*Figure 1G*). This result indicates that the first spacer acquired by a CRISPR-adapted cell determines the spacer content of its progeny during the formation of the colony.

## Priming by the first acquired spacer is likely involved in CRISPR expansion

To analyze in more detail the spacer content of the surviving population, we isolated 26 individual colonies from a replica of the experiment described in *Figure 1C*, amplified their CRISPR array (*Figure 2A*) and subjected the PCR products to next-generation sequencing (NGS). We found 10 colonies that produced > 98% of single-spacer reads (*Figure 2B* and *Figure 2—source data 1*) and therefore they belong to the mono-spacer category, also evident from the results of *Figure 2A*. The remaining 16 colonies displayed a variable fraction of multi-spacer reads, ranging from 5% to 85%. We used these data to determine whether 'priming' is involved in the sequential acquisition of new spacers. During primed CRISPR adaptation, the frequency of spacer acquisition is significantly increased by the presence of a pre-existing targeting spacer (*Datsenko et al., 2012*). When we use our experimental system to investigate priming after infection of cells in liquid culture, the acquisition of additional spacers occurs from regions flanking the Cas9 target site determined by the pre-existing spacer (*Nussenzweig et al., 2019*). To test for primed acquisition in solid media, we extracted 1711 spacer pairs from the NGS data and measured the first-to-second spacer distance (*Figure 2C* and *Figure 2—source data 1*). We found that, similarly to our previous results in liquid cultures, there is a marked enrichment of second spacers derived from the 500 bp region adjacent

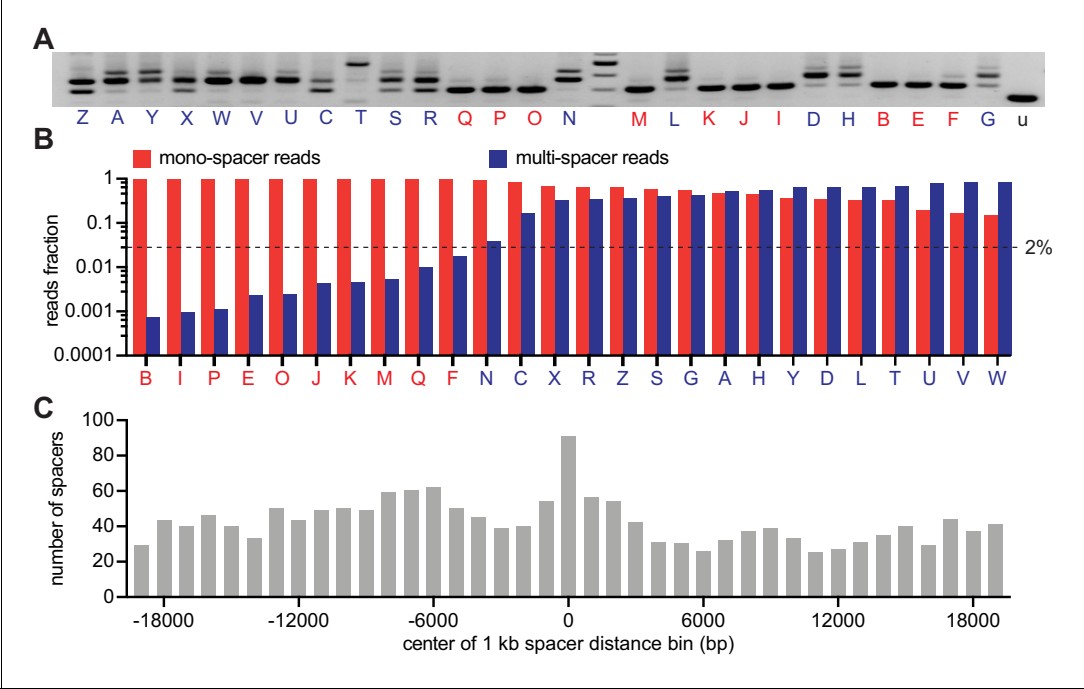

**Figure 2.** CRISPR expansion likely involves priming by the first acquired spacer. (**A**) Agarose gel electrophoresis of the products of the amplification of the pCRISPR array present in staphylococci infected in semi-solid media with φNM4γ4, using plasmid DNA templates extracted from 26 different surviving colonies and the unexpanded array (u); red and blue letters: mono- and multi-spacer colony names. (**B**) Fraction of the reads obtained after NGS of the PCR products shown in (**A**) containing either a single or multiple spacers (red and blue bars, respectively). Dashed line indicates the 2% value, the minimal fraction of multi-spacer reads that leads to multiple PCR products after amplification of the pCRISPR array. (**C**) Distance between the targets in the φNM4γ4 genome specified by the first and second spacers acquired after infection of staphylococci carrying pCRISPR; obtained from analysis of NGS data. The number of different second spacers within 1 kb bins of the φNM4γ4 genome are shown; the position of first spacer acquired in each array is set as 0 kb.

The online version of this article includes the following source data for figure 2:

**Source data 1.** Spacer sequences, their number of reads and their position in the phage genome; obtained from next-generation sequencing of the 26 colonies reported in *Figure 2B*.

to the target of the first spacer. This result suggests that primed, or cleavage-mediated, spacer acquisition plays an important role to create multi-spacer cells and colonies.

## The ability of phage to escape targeting by the founding spacer determines colony heterogeneity

Interestingly, the above results showed that mono-spacer colonies also contained a low proportion of cells harboring multiple spacers, that was not detectable by PCR amplification of the CRISPR array of individual cells from these colonies. This finding demonstrates that multiple spacer acquisition occurs in both colony types, most likely through priming as shown above. Thus, the generation of multi-spacer colonies can be explained by at least two non-exclusive mechanisms. One possibility is that the founder spacer mediates very efficient priming that leads to the generation of high numbers of multi-spacer cells as the colony forms. Another scenario is that additional spacer acquisition through priming mediated by the founder spacer is equally infrequent, but the viral population contains a high frequency of escaper phages that can evade the immunity mediated by the founder spacer. This results in the positive selection of the few cells that contain an extended CRISPR array, leading to the formation of multi-spacer colonies. In this scenario, founder bacteria that acquired spacers with a low escape rate in the viral population grow largely unchallenged; cells harboring additional spacers are not selected and a mono-spacer colony is formed. We tested this hypothesis with three experiments. First, we measured the escape rate of different founder spacers by calculating the fraction of escapers within the phage population. We found that 8/9 founder spacers from multi-spacer colonies showed high rates of escape, where 1 in $10^6$–$10^7$ phages were able to evade

type II CRISPR-Cas immunity (*Figure 3A*). In contrast, founder spacers (10 tested) of mono-spacer colonies displayed a much lower frequency of escape, ranging from undetectable to a maximum of $10^{-8}$. Sequencing of the different targets confirmed that the escaper phages contained seed or PAM mutations (*Figure 3—figure supplement 1*). Second, we looked for the presence of escaper phage in the colonies that resulted from infection of mono-spacer founder cells in the experiment shown in *Figure 1F* and were unable to detect phages that could bypass spacer $F_0$ immunity (*Figure 3—figure supplement 2A*). Third, we measured the presence of phages that specifically evade targeting by the founder spacer within the surviving colony. To do this, we resuspended two multi-spacer phage-resistant colonies in media and centrifuged them to separate cells from soluble phage particles. The pelleted staphylococci were re-streaked and checked for the number of acquired spacers to isolate cells with only one, two or three spacers. Lawns of these bacteria were infected with 10-fold dilutions of phage from either the wild-type stock or isolated from each of the colonies (*Figure 3B* and *Figure 1—source data 1*). We found that staphylococci harboring only the founder spacer were readily infected by the phage isolated from their own colony, but not by the second phage. The acquisition of an additional spacer drastically reduced the propagation of the phage, and the insertion of the third spacer drove plaque formation below the limit of detection of the assay.

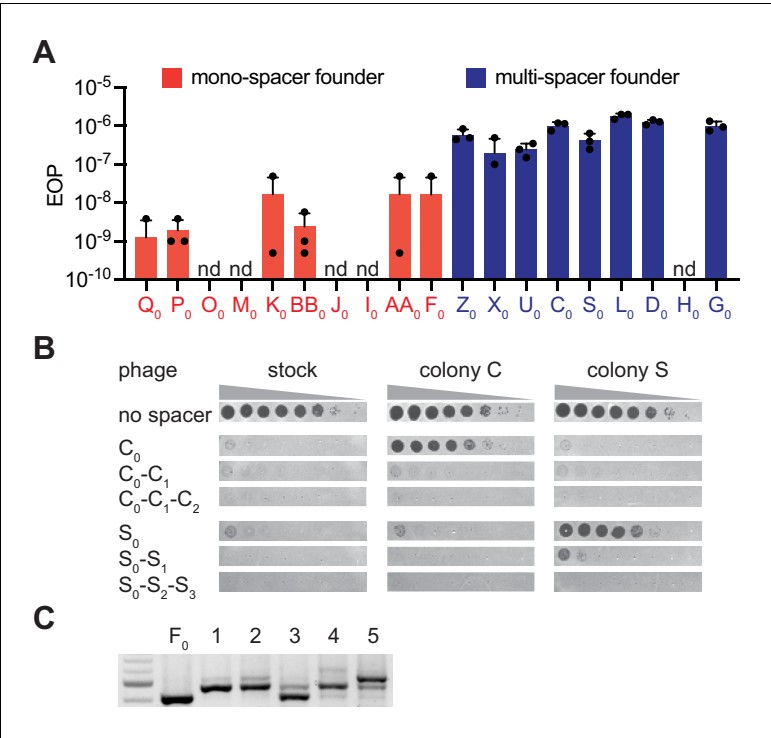

**Figure 3.** The ability of phage to escape targeting by the founding spacer determines colony heterogeneity. (**A**) Efficiency of plaquing (EOP), calculated as the number of φNM4γ4 plaques on the test strain relative to the total number of phage particles in the stock. Different mono-spacer (red) and multi-spacer (blue, carrying only the founder spacer without additional ones) strains were tested. Mean ±STD of thee biological replicates (black dots) are reported. (**B**) Detection of plaques present in 10-fold serial dilutions of either φNM4γ4 phage stock or phage isolated from multi-spacer colonies founded by spacers C or S, spotted on lawns of non-CRISPR staphylococci ('no spacer') or carrying pCRISPR plasmids with an increasing spacer content. Mean ± StDev values of three independent experiments (black dots) are shown; n.d., not detected. (**C**) Agarose gel electrophoresis of the products of the amplification of the pCRISPR(spacer F) array present in five colonies that survived infection in semi-solid media with φNM4γ4 and spacer F escaper phage.

The online version of this article includes the following figure supplement(s) for figure 3:

**Figure supplement 1.** Target sequences of different escaper phages.
**Figure supplement 2.** Quantification of escaper phage within spacer F founder colonies.

Next, we tested if the high proportion of escapers of a founder spacer is the cause for the formation of multi-spacer colonies. If this is the case, founder spacers from mono-spacer colonies would also form multi-spacer colonies when escaper phage are artificially added to the phage stock. We isolated a phage that can overcome targeting by spacer $F_0$ (*Figure 3—figure supplement 1*), a mono-spacer colony founder, and repeated the experiment of *Figure 1F* infecting with a mixture of wild-type and escaper phage. We previously measured that the frequency of escape of spacer $F_0$ is $\sim 10^{-8}$ (*Figure 3A*); we added escaper phage to raise this rate to $10^{-5}$. As opposed to the previous results, the increase in the proportion of escaper phage generated multi-spacer colonies from this otherwise mono-spacer colony founder (*Figure 3C*) and contained phage that escaped the immunity provided by the $F_0$ spacer (*Figure 3—figure supplement 2B*). Together, these results demonstrate that the escape rate of the founder spacer determines the spacer content of the colony, with high levels of escape leading to the formation of multi-spacer colonies.

## Multi-spacer colonies emerge late

Given that primed spacer acquisition is a relatively rare event (*Nussenzweig et al., 2019*), the majority of cells in a developing colony contain only the founder spacer. Therefore, a high rate of escape of this spacer would lead to the predation of most cells in the bacterial community by target mutant phages, possibly even abrogating the formation of the colony. To corroborate this, we quantified the number of colonies obtained in the experiments shown in *Figures 1F, G* and *3C*. Indeed, whereas about 1000 of the 2000 staphylococci harboring the mono-spacer founder $F_0$ added to the top agar survived infection, only about 500 cells containing the multi-spacer founder $C_0$ were able to form colonies (*Figure 4A*). To investigate the role of spacer acquisition in the formation of the colonies, we also infected cells containing pCRISPR($\Delta cas2$), which are unable to acquire new spacers (*Heler et al., 2015*). As expected, this mutation profoundly affected the formation of the $C_0$-founded colonies, which requires in the addition of new spacers to survive escaper phages. In contrast, the $\Delta cas2$ genetic background did not have a big impact in the rise of $F_0$-founded colonies (*Figure 4A*). However, when the $F_0$ cells where infected in the conditions described for the experiment in *Figure 3C*, that is with a phage population carrying a high proportion of escapers that evade the immunity provided by this spacer, colony formation mimicked the results obtained for the $C_0$ founder cells infected with a regular phage stock (*Figure 4A*). These observations confirm our previous results and highlight the role of escaper phage in the formation of different types of CRISPR immune colonies.

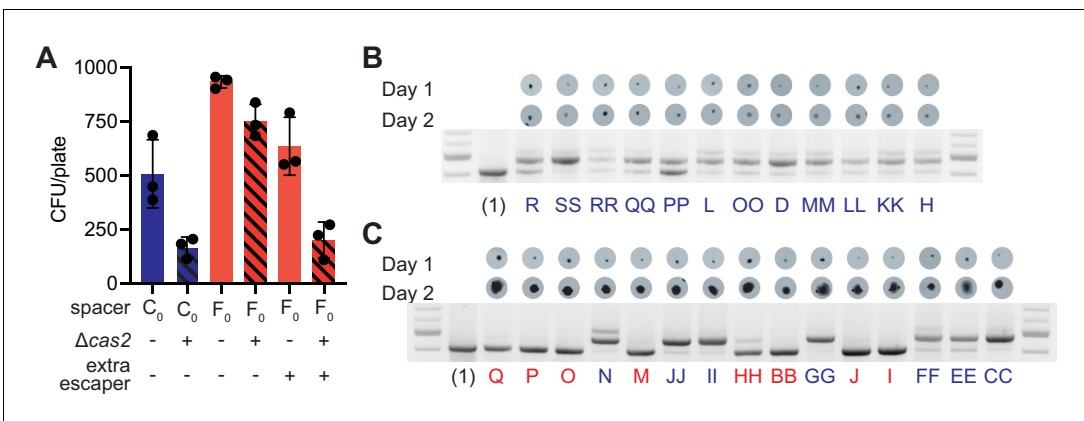

**Figure 4.** Multi-spacer colonies have impaired growth. (**A**) Enumeration of colony forming units (CFU) obtained after infection of staphylococci carrying pCRISPR or pCRISPR($\Delta cas2$) plasmids (clear or dashed pattern bars), containing spacer C or F (red or blue bars, respectively), with φNM4γ4 or φNM4γ4 also containing spacer F escaper phage. Mean ± StDev values of three independent experiments (black dots) are shown. (**B**) Images of bacteriophage-resistant colonies 1 or 2 days after infection with φNM4γ4 phage, and PCR analysis of the spacer content in their pCRISPR plasmid. (*Barrangou et al., 2007*), one spacer control. (**C**) Same as (**B**) but with colonies that experienced growth over time; their letter name colored according to the spacer content: red, mono-spacer; blue, multi-spacer.

While many of the staphylococci that acquire a weak spacer cannot develop a colony, for those that do form a colony it is expected that escaper phages would decimate most of the cells that harbor the founder spacer and only the multi-spacer cells should be able to grow. If this is true, multi-spacer colonies should emerge later than mono-spacer ones. To test this, we performed the experiment of *Figure 1C* and imaged plates at 24 hr and 48 hr after phage infection. We followed 27 colonies displaying similar sizes at 24 hr and found that 12 did not change in size (*Figure 4B*) and 15 that were larger (*Figure 4C*) at 48 hr. Each colony was then checked for the expansion of the CRISPR locus. As hypothesized, all colonies that were unable to grow in size (12/12) acquired multiple spacers (*Figure 4B* and *Figure 1—source data 1*). On the other hand, colonies that increased in size over time contained both a single (8/15) or multiple (7/15) spacers (*Figure 4C* and *Figure 1—source data 1*).

## Multi-spacer colonies display sectored morphologies

We decided to examine in more detail the mono- and multi-spacer colonies that developed a large size at 48 hr (*Figure 4C*). When such colonies were photographed at higher magnification, we found two types of morphologies: smooth and sectored (*Figure 5—figure supplement 1A*). Analysis of the spacer content of several of these colonies revealed that while the smooth colonies contained a single spacer (*Figure 5A*, *Figure 5—figure supplement 1B–D* and *Figure 1—source data 1*), the sectored ones harbored multiple spacers (*Figure 5B*, *Figure 5—figure supplement 1E–G* and *Figure 1—source data 1*). Moreover, analysis of isolated sectors showed that they contained different multi-spacer families with the same founder spacer (*Figure 5B* and *Figure 5—figure supplement 1G* and *Figure 1—source data 1*). To investigate the role of escaper phage in the formation of sectored colonies, we performed the experiment of *Figure 3C*, infecting mono-spacer founder F cells, both capable (wild-type) or unable to acquire new spacers (Δ*cas2*), with an excess of escaper phage. Indeed, exposure to high concentrations of escaper phage, but not to the wild-type phage stock alone, resulted in the formation of sectored colonies (*Figure 5C* and *Figure 5—figure supplement 2A*). Moreover, in the absence of spacer acquisition the addition of escaper phage resulted in the formation of translucent Δ*cas2* colonies, most likely formed before the rise of escapers and then disintegrated. To corroborate these results, we grew mono-spacer, smooth, colonies in the absence of phage for 24 hr and then added an excess of escaper phage on top of them (*Figure 5D*). We also added PBS as a negative control or lysostaphin, a peptidoglycan hydrolase (*Schindler and Schuhardt, 1964*; *Figure 5—figure supplement 2B*), as control for colony lysis. After 24 hr of incubation, we observed that PBS did not alter the morphology of the colonies; that is they remained smooth. As expected, lysostaphin lysed the cells and produced translucent colonies. In contrast, the addition of exogenous escaper phage altered the morphology of the colonies, from smooth to sectored. These results demonstrate that founder spacers determine the morphology of the colony they originate, most likely by affecting the co-evolution dynamics between CRISPR-adapted hosts and CRISPR-escaping phages.

## Host-phage co-evolution within multi-spacer colonies leads to increased resistance

The model described above suggests that cells within the multi-spacer colonies, originally founded by a weak spacer, co-evolve over time with the phage to limit its ability to infect. Strong founder spacers, in contrast, should quickly neutralize the phage. To test these hypotheses, we performed a time-shift experiment that evaluates the resistance of bacterial samples over time against phage from past, concurrent, and future time points (*Common et al., 2019*; *Laanto et al., 2017*). We repeated the experimental setup used in *Figure 1F* (mono-spacer founder F) and 1G (multi-spacer founder C), collecting 12 surviving colonies at 24 hr, 36 hr, and 48 hr post-infection and isolating and amplifying phage from each of them. Finally, each colony, as well as their respective founders, were grown in liquid cultures and used to seed top agar media. On these plates, 2 μl of stock phage (which has not co-evolved with CRISPR-resistant staphylococci and was used to obtain the 'time zero' datapoint) or phage isolated from each colony was spotted. After overnight incubation at 37° C, we evaluated the resistance of the bacteria on the top agar as follows: bacterial growth in the zone where phage was spotted was considered as 'full resistance' and given a score of 1; complete inhibition of growth (clear spot) was determined to be 'no resistance' and given a score of 0. Partial

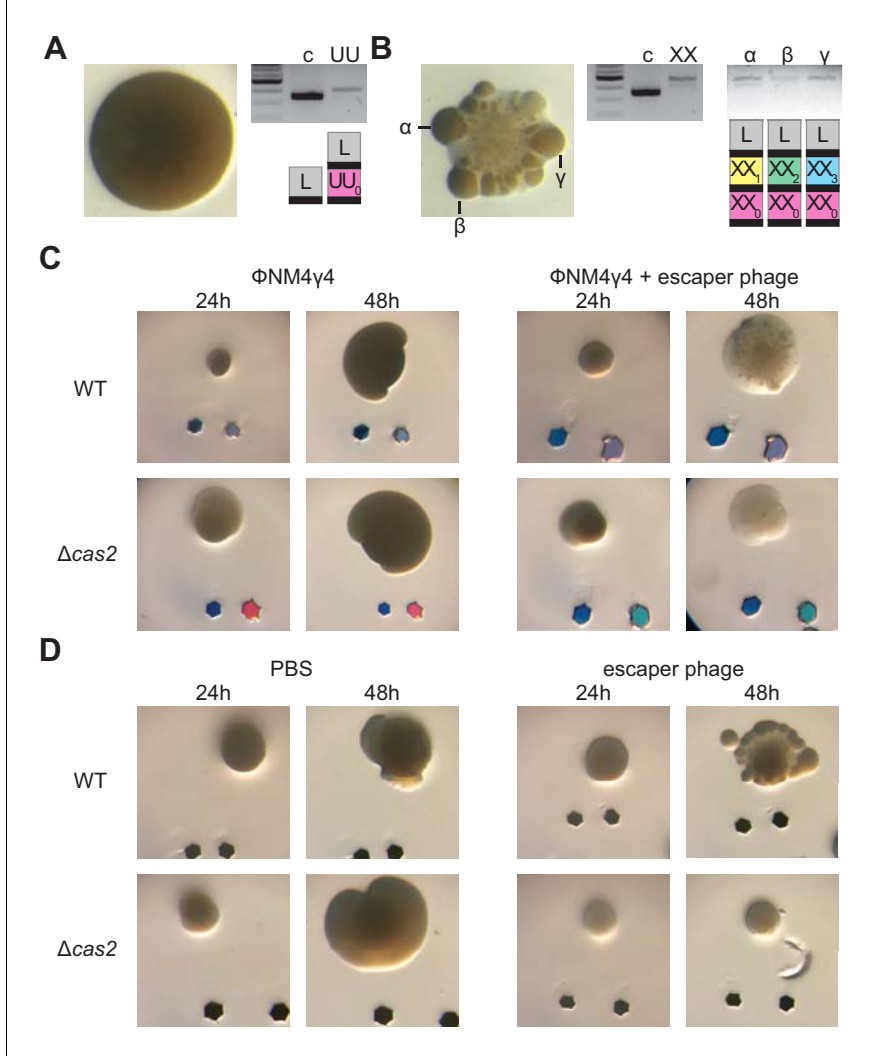

**Figure 5.** Multi-spacer colonies display sectored morphologies. (**A**) Image of a smooth colony (containing the founder spacer UU) as well as the gel agarose analysis of PCR products obtained after amplification of its pCRISPR array; 'c' shows amplification of pCRISPR, a no-spacer control. (**B**) Same as (**A**) but for the sectored colony founded by spacer XX. Also showing the gel agarose analysis of PCR products obtained after amplification of its pCRISPR array present in three different sectors (α,β,γ) with a schematic of the pCRISPR array determined after sequencing of the PCR product (L, leader; black rectangle, repeat; colored boxes, acquired spacers with different colors indicating different spacer sequences within a colony). (**C**) Images of representative colonies grown 24 or 48 hr after top agar infection of wild-type or Δcas2 mono-spacer founder F cells, with ϕNM4γ4 containing or not additional spacer F escaper phage. Glitter markers are shown to normalize both the position as well as the size of the image at different times. (**D**) Images of representative colonies of wild-type or Δcas2 mono-spacer founder F cells grown for 24 hr in the absence of phage, when a drop of either PBS or spacer F escaper phage was added on top and a second image was obtained 24 hr after. Glitter markers are shown to normalize both the position as well as the size of the image at different times.

The online version of this article includes the following figure supplement(s) for figure 5:

**Figure supplement 1.** Analysis of additional smooth and sectored colonies.

**Figure supplement 2.** Requirement of spacer acquisition for the formation of sectored colonies.

inhibition was given a score of 0.5. The resistance scores for the bacteria against these 12 phage populations of a given time-point were averaged for each host, with the exception of the time zero datapoints, obtained with only the stock phage (*Figure 6—source data 1*).

To evaluate whether colonies of a given time-point gained or lost resistance over time, we calculated the average resistance of a colony against all the phage from the same time point and plotted

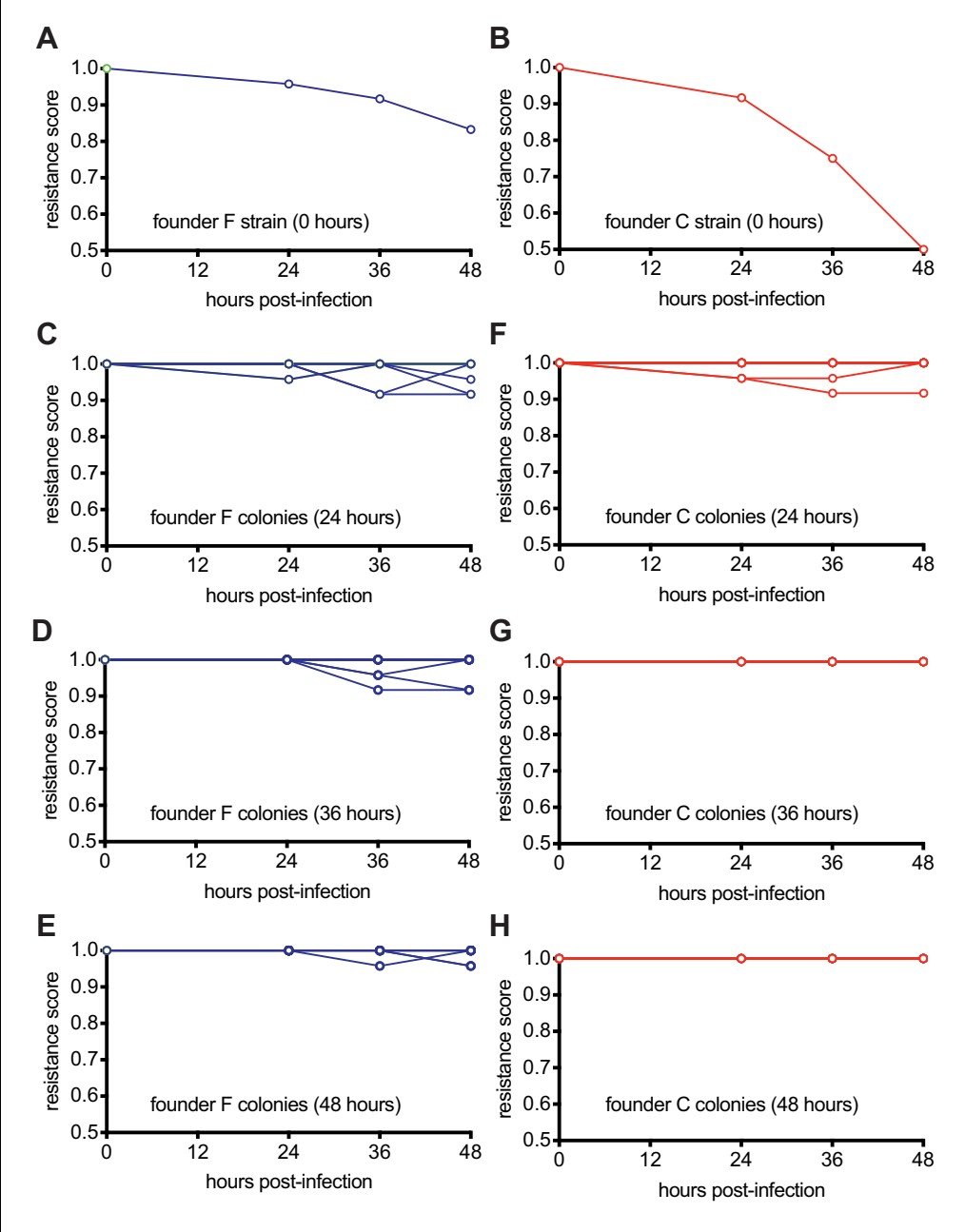

**Figure 6.** Host-phage co-evolution within multi-spacer colonies leads to increased resistance. (A) Resistance score of staphylococci carrying the pCRISPR programmed with founder spacer F to the stock φNM4γ4 phage (time 0) or the phages isolated from colonies that survived 24, 36 and 48 hr after infection. (B) Same as (A) but for cells with pCRISPR programmed with founder spacer C. (C–E) Same as (A) but showing the resistance scores for 12 different colonies isolated at 24 (B), 36 (C) and 48 hr (E) after infection. To the stock φNM4γ4 phage (time 0) or the phages isolated from colonies that survived 24, 36 and 48 hr after infection. (F–H) Same as (C–E) but for cells with pCRISPR programmed with founder spacer C.
The online version of this article includes the following source data for figure 6:

**Source data 1.** Raw data used for the plots shown in *Figure 6*.

the score averages for each colony over time (*Figure 6* and *Figure 6—source data 1*). We found that the mono-spacer founder F is initially resistant to the stock phage, but experiences a small decrease in immunity (from a score of 1 to 0.83) against all 'future' phages (*Figure 6A*),

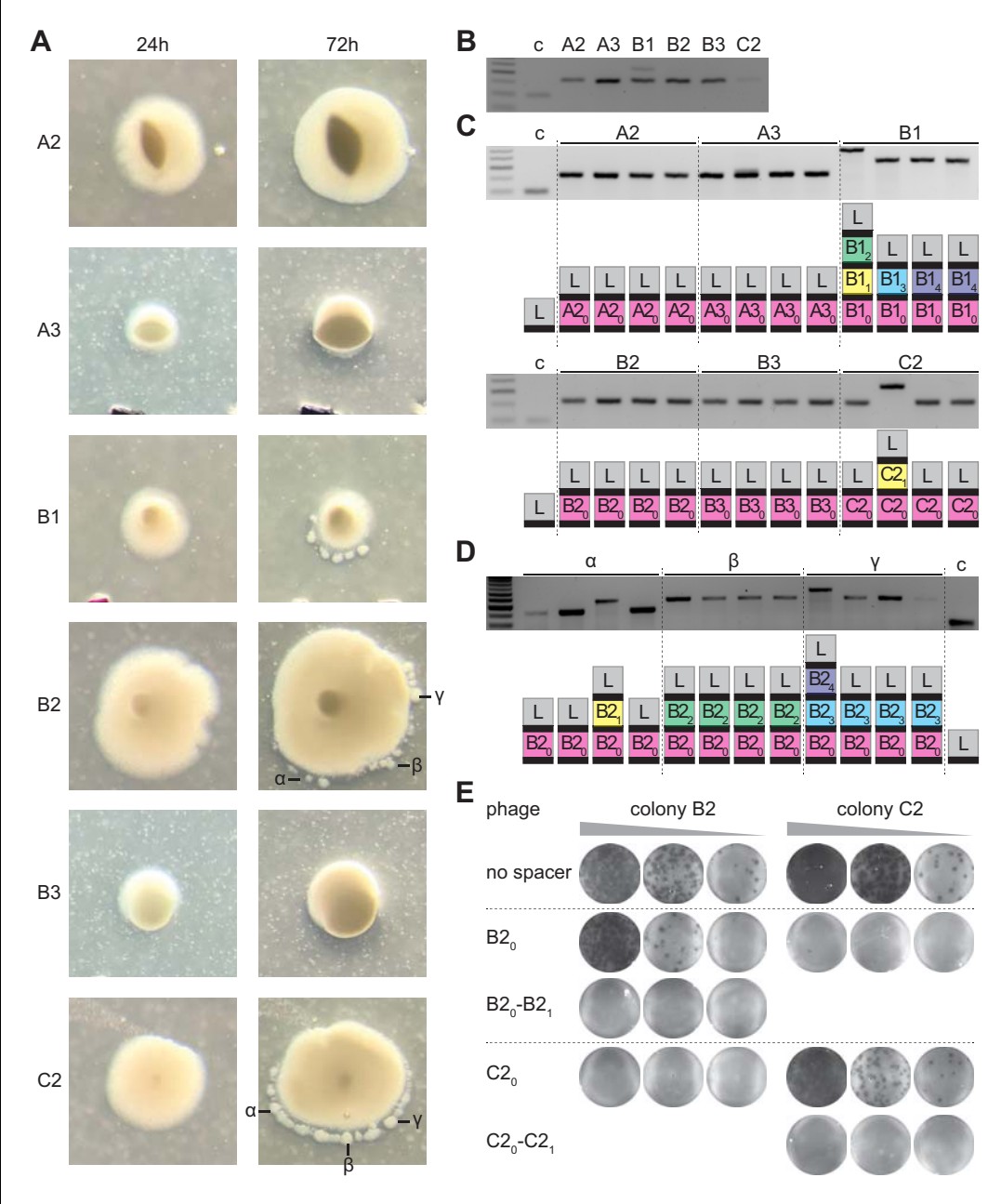

**Figure 7.** Analysis of *S. thermophilus* phage-resistant colonies. (**A**) Phage-resistant colonies obtained 24 or 72 hr after infection of *S. thermophilus* DGCC7710 with ϕ2972. Satellite colonies further analyzed are labeled (α, β and γ). (**B**) Agarose gel electrophoresis of the products of the amplification of the *S. thermophilus* CRISPR1 locus, using DNA templates extracted from the center of the colonies shown in (**A**); 'c' shows amplification of pCRISPR, a no-spacer control. (**C**) Colonies resulting from the re-streak of the colonies shown in (**A**), with a schematic of the pCRISPR array determined after sequencing of the PCR product (L, leader; black rectangle, repeat; colored boxes, acquired spacers with different colors indicating different spacer sequences within a colony); 'c' shows amplification of pCRISPR, a no-spacer control. (**D**) same as (**C**) but after re-streak of the satellites from colony B2. (**E**) Detection of plaques present in 10-fold serial dilutions of ϕ2972 phage isolated from multi-spacer colonies founded by spacers B2 or C2, spotted on lawns of streptococci lacking a targeting spacer ('no spacer') or carrying one or two targeting spacers.

demonstrating that the co-evolution of the phage with cells carrying this spacer generates a low number of escapers that increases over time. This co-evolution also increases the resistance of post-infection bacteria collected at 24, 36 and 48 hr (*Figure 6B–C*), leading to a stable co-existence of phage and resistant bacteria at the end of the experiment. In contrast, the results obtained after following the infection of spacer C hosts revealed more drastic co-evolution dynamics. First, founder cells showed a marked susceptibility to 'future' phages that co-evolved over time (from a score of 1 to 0.5, *Figure 6E*), an observation in agreement with previous results showing that spacer C is more likely to encounter escaper phages (*Figure 3A*). Second, co-evolution increases the overall resistance of colonies collected at 24 hr after infection (*Figure 6F*) and then leads to complete resistance at 36 and 48 hr post-infection (*Figure 6G–H*). Therefore, the increased pressure that founder spacer C faces from escaper phages seems to both accelerate and strengthen the evolution of resistance.

## Multi-spacer acquisition within Streptococcus thermophilus sectored colonies

One caveat of the experiments presented above is that they are performed using an engineered system. We wanted to determine if our findings could generally apply to the co-evolution in solid media of prokaryotes harboring type II CRISPR-Cas systems and their phages. To do this, we explored spacer acquisition in *Streptococcus thermophilus*, the first organism shown to employ CRISPR-Cas systems for anti-phage defense (*Barrangou et al., 2007*). Strain DGCC7710 harbors two type II-A CRISPR loci (CRISPR1 and CRISPR3) that function independently from one another (*Carte et al., 2014*) and although both are capable of spacer acquisition, CRISPR1 is preferentially used (*Barrangou et al., 2007*; *Wei et al., 2015*). We infected naive cells with the phage φ2972 in soft agar to isolate resistant colonies as previously described (*Hynes et al., 2017*). After inspection under the microscope, we detected colonies with or without small satellites (*Figure 7A*), and we selected three of each type for PCR analysis. DNA samples obtained from the center of the colonies showed only one spacer inserted into the CRISPR1 locus except for the satellite-containing colony B1 (*Figure 7B*). As we did before, to analyze the spacer content of different cells within these colonies, we re-streaked them and amplified and sequenced the CRISPR1 locus of four of the resulting colonies. We found that regular colonies were composed of streptococci harboring the same spacer sequence (*Figure 7C* and *Figure 1—source data 1*). The satellite-containing colonies B1 and C2, but not B2, contained cells with 1, 2 or 3 spacers, with a common sequence for the first spacer acquired (*Figure 7C* and *Figure 1—source data 1*). Finally, we isolated DNA from specific satellites to determine their spacer content. We found that colony B2, for which we detected only mono-spacer cells with the central body, contained satellites with 1, 2 or 3 spacers (*Figure 7D* and *Figure 1—source data 1*). A similar outcome was obtained after the analysis of satellites surrounding colony C2 (*Figure 1—source data 1*). Based on the above results with staphylococci, we wondered whether the immunity provided by founder spacer of satellite colonies is easily bypassed by escaper phages present within the colony. To test this, we performed the experiment shown in *Figure 3B*, isolating phage from the B2 and C2 colonies and measuring their propagation on streptococci harboring only the founder spacers for these colonies or an additional spacer (*Figure 7E*). Indeed, we found that the B2 founder spacer provided strong immunity to cells infected with the phage collected from the C2 colony but failed to protect the host from phage isolated from the B2 colony. The additional B2 spacer, on the other hand, restored full immunity. Similarly, no plaques were observed upon infection of the streptococci carrying the C2 founder spacer with phage obtained from the B2 colony, but no protection was conferred by this founder spacer against infection with phage from colony C2, a result that was reverted by the addition of a second spacer. Altogether, these results show that CRISPR-adapted *S. thermophilus* colonies contain satellites with multiple spacers, all harboring the same first, founder spacer. Phages within the colony can escape the immunity provided by the founder spacer, but not the protection conferred by the presence of additional spacers. Although there are differences with the previous results obtained with our heterologous experimental system (that could be explained in part by the use of different organisms and viruses; see Discussion below), these results indicate that the type II CRISPR-Cas immune response similarly affects the development of *S. thermophilus* and *S. aureus*-resistant colonies: the immunological strength of founder spacers determine the morphology of the colony they originate, most likely by affecting the viral-host co-evolution dynamics.

## Discussion

Here, we studied the formation of resistant colonies during the type II CRISPR-Cas immune response. CRISPR immunity starts with the acquisition of a spacer sequence from the invading phage, which confers resistance to the cell. When infection occurs in solid media, each of these resistant bacteria can form a colony. We found that the strength of the immunity provided by the spacer acquired in the founder cell determines the nature of the virus-host co-evolution that occurs during the formation of the colony. This co-evolution in turn defines the spacer diversity, emergence rate, morphology and overall resistance of the colony. Founder cells harboring a weak spacer, which immunity can be overcome by a relatively high number of target mutant phages in the viral population, engage in a dynamic co-evolution with the virus. These cells suffer the predation of escaper phages and struggle to emerge; members of the colony in formation that carry additional spacers are positively selected and their growth leads to the formation of micro-colonies (either within or surrounding the original colony) composed of cellular conglomerates with the increased spacer diversity necessary to limit the propagation of escaper phages. On the other hand, the phage is largely contained by founder cells harboring strong spacers, and little viral-host co-evolution occurs. Escapers do not rise, cell growth is not limited, and smooth, mono-spacer colonies are formed. Interestingly, the initial and final levels of resistance of the colonies are reverted. The virus-host co-evolution mediated by the CRISPR-Cas immune response enables the conversion of a genotype that initially confers a high level of susceptibility to the host (a weak founder spacer) into a new genotype that not only promotes survival from co-evolving phages, but also creates a completely resistant host.

How are multiple spacers acquired within the same cell? In one scenario, as the mutant phages that escape the immunity provided by the founder spacer rise in the colony, the infection of the now susceptible founder cells results in a second event of spacer acquisition. In this case, the accumulation of multiple spacers in the same CRISPR array will require several consecutive episodes of escape and reinfection. Since the fraction of cells that acquire additional spacers is very low, about one cell per $10^7$ to $10^6$ in our *S. aureus* experimental system (*Heler et al., 2017*), the low number of cells in a developing colony would make it mathematically impossible for naive CRISPR adaptation to occur three or four consecutive times. A second, more plausible, scenario involves the 'primed' acquisition of multiple spacers during a single infection event. After the initial insertion of the founder spacer, cells contain Cas9 molecules programmed to cleave the viral genome. This cleavage generates double-strand DNA breaks that not only protects the host, but also generates the free DNA ends from which new spacers are acquired. The result is the acquisition of additional spacers from the vicinity of the target site specified by the founder spacer and allow the host to anticipate the rise of escapers (*Nussenzweig et al., 2019*). The high proportion of second spacers that are located within 1 kb of the founder spacer target site (*Figure 2C*) suggest that this is the preferred mechanism, although more detailed work will have to be carried out in the future to decisively understand how multiple spacers are acquired within one colony. In contrast to the infrequent and sequential 'naive' spacer acquisition from escaper phages, 'primed' spacer acquisition can lead to the addition of several spacers within a single infection event. We think that this can occur in a small fraction of the cells, early during colony formation. These multi-spacer cells are amplified or not throughout the development of the colony, depending on the abundance of escaper phage that evade the founder spacer.

Interestingly, the most detailed analysis performed in this study, that is the detection of expanded CRISPR arrays via next generation sequencing, showed that the fraction of multi-spacer loci within each colony is highly variable (*Figure 2B*). There are two non-mutually exclusive factors that could affect this value: the escape frequency and/or the priming efficiency of the founder spacer. We believe that the phage escape frequency of the founder spacer impacts more profoundly the fraction of multi-spacer cells in colonies that, when analyzed with less powerful methods such as colony PCR (*Figure 2A*) and microscopy (*Figure 5A–B*), display a multi-spacer and sectored phenotype. This is because high efficiency of priming would lead to the immediate acquisition of multiple spacers by a high proportion of the infected cells early during the development of the colony. This should result in the formation of smooth multi-spacer colonies, as it would achieve an early neutralization of the founder spacer escapers and allow the unchallenged emergence of the colony. Therefore, we believe that the wide distribution of the fraction of multi-spacer cells within sectored colonies reflects the different escape frequencies for each founder spacer. While we defined founder spacers as weak or strong, in truth the escape frequencies have many different values depending on

the region of the phage genome targeted by each spacer (*Chabas et al., 2019*). At one end of the spectrum, there will be the spacers that target viral protospacers for which any modification of the seed or PAM sequence would lead to complete loss of phage viability. At the other end will be the spacers that match a protospacer that when their seed or PAM sequences are changed, the mutation enhances viral propagation. Many escaper mutations, however, will fall in between these two scenarios, and will include mutations that only decrease or are neutral for the fitness of the phage. This will produce a wide range of escaper mutation frequencies in the viral population that will lead to different levels of selection and accumulation of multi-spacer cells within sectored colonies. We looked for a correlation between the mono-spacer founders and gene function. However, we could not find an enrichment of essential genes targeted by these spacers, probably because even within essential genes there are non-critical amino acids that can be mutated without any fitness effect on the phage. Future work using deep sequencing to detect both all possible phage escape mutations and acquired spacers could shed light on this issue.

On the other hand, the priming efficiency of the founder spacer should be determinant for the different proportions of multi-spacer cells in colonies that appear as mono-spacer and smooth. While founder spacers of smooth colonies also display different escape frequencies (*Figure 3A*), these are not high enough to affect the growth of the cells within the colony (*Figure 4C*). Once cells reach stationary phase, they can no longer be infected by the escaper phage (bacteria in this growth stage are commonly refractory to phage infection) and therefore they cannot propagate to promote the expansion the small fraction of multi-spacer cells present in smooth colonies. Instead, low levels of escaper phage remain in these colonies, even after 48 hr of the initial infection (*Figure 6E*). We determined an empirical value of 2% as the maximum fraction of multi-spacer array that can produce a mono-spacer and smooth phenotype (*Figure 2A–B*). Since the multi-spacer members of these colonies are not important to limit the propagation of the escaper phages and are not subject to strong positive selection during colony formation, their fraction is most likely a result of the different priming efficiencies of their founder spacers.

We performed our studies with two different experimental type II-A CRISPR-Cas systems: *S. aureus* cells harboring the *S. pyogenes* locus and *S. thermophilus*, containing its own locus. While results were similar, that is multi-spacer colonies contained additional sectors or satellites as well as high titers of phage able to escape the immunity provided by the founder spacer, there were also interesting differences. First, infection of liquid cultures produced apparently opposite results. For the heterologous system, this experiment led to a multi-spacer population composed of mono-spacer cells (*Heler et al., 2015*; *Heler et al., 2019*). This is because the initial spacer diversity neutralizes most escape variants present in the viral population and allows the fast growth of the culture. When it reaches stationary phase, it becomes immune to further infection and viral-host co-evolution stops. In contrast, previous experiments with *S. thermophilus* DGCC7710, in which the infected liquid cultures are artificially diluted before cells can reach stationary phase, resulted in the accumulation of multi-spacer CRISPR loci in the population. (*Common et al., 2019*; *Levin et al., 2013*; *Paez-Espino et al., 2015*). It has been shown that under a dilution regiment, *S. thermophilus* and its phage ϕ2972 both increase their resistance and infectivity, respectively, a result that demonstrates the existence of a co-evolutionary arms race (*Common et al., 2019*). After 9 days of the initial infection phages lose their infectivity and the surviving cells contain multiple spacers. However, we believe that these divergent results are the consequence of the lack of dilution in our experiments with the *S. aureus* heterologous system. This is because dilution (i) prevents the culture from accumulating phage-resistant stationary phase cells, and (ii) reduces the spacer diversity of the culture and leads to the selection of multi-spacer that can contain the escapers.

A second disparity is the phenotype and proportion of the multi-spacer colonies:~50% of *S. aureus* colonies display different sectors and ~10% of *S. thermophilus* colonies are surrounded by small satellites. We speculate that this could be the result of differences in growth between these organisms, as well as differences in the ability of the ϕNM4γ4 and ϕ2972 phages to disseminate within forming colonies. Streptococcal cells may form tight communities that cannot be penetrated by ϕ2972, restricting phage predation to the outskirts of the colony. If so, the bulk of the viral-host co-evolution will occur in this space, and selection of multi-spacer survivors will lead to the formation of satellite colonies. On the other hand, *S. aureus* growth may happen in a way that does not shield the internal cells from ϕNM4γ4 predation, leading to selection of multi-spacer members of the community within the whole colony. This model is supported by the opposite outcomes of the initial

amplification of the CRISPR locus. In order to preserve the colonies for further analysis (to determine the spacer content at different times or from different parts of the colony, or to isolate phages), we scooped a small fraction of the center of the colony to extract DNA. Although we were able to detect multiple spacers in the *S. aureus* cells obtained in this way (*Figure 1C*), we did not in the case of *S. thermophilus* cells, even in those derived from colonies with satellites, which prove to contain multiple spacers upon more detailed analysis (*Figure 7B*).

In summary, we found that the acquisition of multiple spacers within a single CRISPR array compensates for the inefficient defense provided by an initial weak spacer, in conditions of structured growth where this compensation cannot be offered by a strong neighbor cell. Therefore, the type II CRISPR-Cas immune response is able to provide robust immunity through the generation of either a diverse population composed of weak and strong cells or a diverse cell harboring weak and strong spacers.

## Materials and methods

### Bacterial strains and growth conditions

Cultivation of *S. aureus* RN4220 (*Kreiswirth et al., 1983*) or OS2 (*Schneewind et al., 1992*) was carried out in tryptic soy broth (TSB) at 37˚C. *S. aureus* media was supplemented with chloramphenicol at 10 µg/ml to maintain pCRISPR plasmids. *S. thermophilus* DGCC7710 was grown in LM17 media at 42˚C.

### Spacer acquisition in liquid cultures

Overnight cultures of *S. aureus* RN4220 containing pCRISPR or pCRISPR($\Delta$cas2) (pWJ40 or pRH61, respectively *Heler et al., 2015*) were diluted 1:100 in 20 mL of BHI and grown for 4 hr shaking at 37˚C. The optical density value at 600 nm ($OD_{600}$) was measured and used to calculate the colony forming units (CFU) per µL. For the infection, 1.3 billion CFU of pCRISPR- or pCRISPR($\Delta$cas2)-containing cells were mixed with 2.6 billion plaque forming units (PFU) of $\phi$NM4$\gamma$4 bacteriophage (*Goldberg et al., 2014*) for a starting MOI of 2. This mixture was added to 65 mL of 50% Heart Infusion Broth (HIB) supplemented with 5 mM $CaCl_2$ and incubated under shaking conditions at 37˚C for 48 hr. Cells were stored by collecting 1 ml of the culture, which was mixed with 50% glycerol and stored at 4˚C or frozen at −20˚C. Individual colonies were isolated by spreading the culture on a TSA plate.

### Spacer acquisition in semi-solid media (top agar)

Infections in top agar were performed with almost the same conditions as those in liquid culture, except that the phage mixture was added to 5 mL of melted 50% Heart Infusion Agar (HIA top agar) supplemented with 5 mM $CaCl_2$ instead of added to liquid media. The top agar was then poured onto a plate containing solidified Tryptic Soy Agar (TSA). Plates were then incubated at 37˚C for 48 hr. Each colony was picked and resuspended in 30 µL of TSB and stored at 4˚C or frozen at −80˚C with 50% glycerol. To isolate individual cells from the colony, the adapted colony was diluted in TSB and streaked on TSA plates.

Spacer acquisition in *S. thermophilus* cells was carried out with lytic phage 2972 according to a previously developed method (*Hynes et al., 2017*).

### Simulation of spacer acquisition in semi-solid media

To recreate the infection conditions encountered by the founder cell (an isolated CRISPR-immune cell suffering infection by a very high number of phages; the result of their exponential propagation within sensitive hosts that were not able to acquire spacers), we mixed 2000 CFU of the founder cells (harboring pCRISPR with the addition of the founder spacer) with 1.3 billion CFU of pCRISPR($\Delta$cas2) cells before adding phage and mixing with top agar.

### CRISPR array amplification

To evaluate the size of the CRISPR array, we mixed 2 µl of the resuspended colony to 20 µL of colony lysis buffer with 50 ng/µl lysostaphin (*Heler et al., 2015*). This mixture was boiled at 98˚C for 10 min and cooled down at 37˚C for 20 min to lyse the cells. To amplify the CRISPR array, we used 1 µl

of the supernatant as template for a TopTaq PCR amplification (Qiagen) with primers H54 and H237 (see *Supplementary file 1*). The resultant PCR amplicons were then analyzed on 2% agarose gels. To further characterize individual sister cells within the colony, these PCR products were submitted for Sanger sequencing.

The CRISPR1 locus of *S. thermophilus* was amplified as follows: the whole colony or 2 µl of resuspended colony were added to 20 µL of colony lysis buffer and incubated at 98°C for 10 min. To amplify the CRISPR1 array we added 0.5 µl of the supernatant as template for a TopTaq PCR amplification (Qiagen) with primers NA19 and NA20.

Next-generation sequencing pCRISPR plasmids were isolated from 10 µL of the frozen colonies with modified QIAprep Spin Miniprep Kit protocol as previously described (*Modell et al., 2017*). We used 2 µL of the plasmid preparation as template for PCR amplification with Phusion DNA Polymerase (Thermo) using primers NP389 and NP390 (see *Supplementary file 1*). Amplicons were given to the Rockefeller University Genomics Core for library preparation and Illumina Hi-Seq Sequencing. The data was analyzed using Python: spacer sequences were extracted ordered in according to their position within the array and aligned to the ϕNM4γ4 reference genome.

## Quantification of phages escapers for different founder spacers

Top agar lawns of each founder cell were made by mixing 100 µL of an overnight culture with 5 mL of HIA top agar supplemented with 5 mM $CaCl_2$. We then plated 2 µL of a serially diluted phage stock on the lawn and incubated the plate at 37°C. After 24 hr we were able to count PFU and calculate the escaper rate for a particular founder spacer as the ratio to the total PFU count of the phage stock. DNA from individual escaper plaques was isolated according to previously described techniques (*Goldberg et al., 2014*). The target region of each founder spacer was then amplified using the following oligonucleotides (see *Supplementary file 1*): Founder Z: H463, H493, Founder X: H40, NP255, Founder G: NP170, AV363, Founder S: NP331, H462, Founder D: NP180, H485, Founder Q: W1051, NP182, Founder P: NP279, W1087, Founder K: H501, H471, Founder BB: H122, H135, Founder F: AV462, H453, Founder AA: H477, NP313, Founder Y: NP265, H613, Founder C: H122, H135, Founder L: H501, H471; and the PCR products sent for Sanger sequencing.

For streptococcus, lawns of each strain were made by mixing 300 µL of an overnight culture with 3 mL of LM17 top agar supplemented with 10 mM $CaCl_2$. We then added 40 µL of a serially diluted phage stock and incubated the plate at 42°C.

## Isolation of phage from colonies

To propagate the phage and increase its titer, 10 µL of the supernatant from the resuspension of a resistant colony were mixed with 100 µL of an overnight culture of *S. aureus* OS2, an erythromycin-resistant non-immune strain, and 5 mL of HIA top agar supplemented with 5 mM $CaCl_2$ instead. The mix was plated on erythromycin-containing TSA plates to ensure that host bacteria were killed. After 24 hr of incubation at 37°C phages were purified by filtration as previously described (*Heler et al., 2015*).

To isolate and make a stock of phages from *S. thermophilus* colonies, 10 µL of the supernatant from the resuspension of a resistant colony were mixed with 10 ml LM17 with 10 mM $CaCl_2$ and a volume corresponding to 0.1 $OD_{600}$ value of *S. thermophilus* JAV27 cells (*Varble et al., 2019*) which type II CRISPR-Cas loci. After 4 hr incubation at 42°C, the cultures were sterile filtered to collect the amplified phage particles.

## Challenge of founder spacer F cells with additional escaper phage

First, an escaper phage containing mutations in both the seed and PAM sequences of the spacer F target was isolated (*Figure 3—figure supplement 1*) by plaquing ϕNM4γ4 on spacer F cells and sequencing the target to determine the escape mutations. The escaper was propagated on non-CRISPR staphylococci (*S. aureus* OS2) to high titers. Finally, 20,000 PFU of the escaper stock were added to the liquid HIA top agar mixture during the spacer acquisition assay described above.

## Determination of colony size over time

Images of the plates with ongoing spacer acquisition assays at 24 hr and 48 hr were acquired with a GE ImageQuant LAS 4000 Imager at a 1X magnification, and aligned using Adobe Illustrator.

## Microscopy analysis of colony morphology

Colonies on spacer acquisition plates were marked with pieces of glitter of different colors, which also served as a size marker. Images were taken with a 10X magnification with a Nikon SMZ-800N Stereo Zoom Microscope.

Construction of pCRISPR(spacer F, Δcas2) *cas2* was deleted from pCRISPR(spacer F) following the protocol used to convert pWJ40 into pRH61 (*Heler et al., 2015*).

## Treatment of colonies

0.5 µl of either 1X Phosphate Buffer Solution (PBS), 5 mM lysostaphin, or spacer F escaper phage stock were added on top of different colonies growing on plates.

## Time-shift assay

To measure the resistance of the bacteria to phage over time, top agar infection assays using founder F or C cells were performed as in *Figure 1F* or 1G, respectively. Samples were collected at 24, 36 and 48 hr post-infection by resuspending 12 of the resulting colonies in TSB, except for the experiment with founder F cells at 36 hr, for which 11 samples were recovered.

Phage from each sample was isolated and amplified as follows. First, 2 µL of each colony supernatant was spotted onto the surface of top agar seeded with non-immune *S. aureus* OS2 (for the phage used at time 0, we spotted 12 plaques obtained by plaquing the original stock of ϕNM4γ4 phage on top agar containing non-CRISPR hosts). Second, the phage 'spots' obtained were resuspended into 20 µL of TSB and stored at 4°C.

To test the resistance of the bacteria over time, 2 µl of each colony resuspension were grown overnight in 200 µl of BHI media at 37°C. 150 µl of these cultures were mixed with 5 ml of HIA top agar containing 5 mM of $CaCl_2$ and poured over plated on TSA plates. Time 0 bacteria were directly grown overnight at 37°C in BHI from our stocks of founder F and C cells. 2 µl of each amplified phage sample was spotted on top agar seeded with cells from different colonies (or founders in the case of time 0 infections). Plates were incubated overnight at 37°C and the resistance of the bacteria against the phage was scored as 0 (complete lysis), 0.5 (partial resistance), and 1 (full resistance).

## Acknowledgements

We are grateful to Dr. Edze Westra (Exeter University) for critical reading of our manuscript and for providing thoughtful suggestions. We thank the Rockefeller University Genomics Resource Center for assistance with next generation sequencing experiments. LAM is supported by a Burroughs Wellcome Fund PATH Award, and a NIH Director's Pioneer Award (DP1GM128184). LAM is an investigator of the Howard Hughes Medical Institute.

## Additional information

### Competing interests

Luciano A Marraffini: is a cofounder and Scientific Advisory Board member of Intellia Therapeutics, and a co-founder of Eligo Biosciences. The other author declares that no competing interests exist.

### Funding

| Funder | Grant reference number | Author |
| --- | --- | --- |
| NIH Office of the Director | DP1GM128184 | Luciano A Marraffini |
| Burroughs Wellcome Fund | PATH Award | Luciano A Marraffini |
| Howard Hughes Medical Institute | Investigator | Luciano A Marraffini |

The funders had no role in study design, data collection and interpretation, or the decision to submit the work for publication.

## Author contributions
Nora C Pyenson, Conceptualization, Software, Formal analysis, Investigation, Visualization, Writing - original draft, Writing - review and editing; Luciano A Marraffini, Conceptualization, Supervision, Funding acquisition, Writing - original draft, Writing - review and editing

## Author ORCIDs
Luciano A Marraffini (iD) https://orcid.org/0000-0002-9163-0969

## Decision letter and Author response
Decision letter https://doi.org/10.7554/eLife.53078.sa1
Author response https://doi.org/10.7554/eLife.53078.sa2

# Additional files

## Supplementary files
- Supplementary file 1. List of oligonucleotide primers used in this study.
- Transparent reporting form

## Data availability
NGS data are available in the accompanying source data files.

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
