## [Decision Letter]

**Acceptance summary:**

This study elegantly describes how growth in solid media alters the dynamics of spacer acquisition during phage infection, with profound results for the development of effective immunity. The structured environment of the solid media enables repeated encounters between phage and host, which increases the acquisition of multiple spacers in the Type II CRISPR-Cas system, particularly in cases where the initial spacer was suboptimal. This mode of immunity leads to the emergence of greater spacer diversity in the population, increasing the likelihood of surviving subsequent challenge with escaper phage.

**Decision letter after peer review:**

Thank you for submitting your article "Co-evolution within structured bacterial communities results in multiple expansion of CRISPR loci and enhanced immunity" for consideration by *eLife*. Your article has now been reviewed by Aleksandra Walczak as the senior editor, myself acting as the Reviewing Editor, and three reviewers. Joe Bondy-Denomy (Reviewer #1) has agreed to reveal his identity.

Overall, the reviewers were enthusiastic about your work describing the acquisition of bacterial immunity against phage in structured environments. The question addressed by the study was deemed novel and the major finding – that the first spacer acquired dictates the course of subsequent immune adaptation – was regarded novel and interesting. However, all three reviewers agreed that the manuscript was difficult to read, and the data were at times presented in a confusing manner. Adding schematics to represent the relationship between colonies or the order of experimentation would be one way of improving the readability of the article. To make the article accessible to the broad readership of *eLife*, we therefore request that the article be edited in accordance with the comments below before it can be accepted for publication.

Summary:

CRISPR-mediated adaptive immunity relies on the acquisition of spacers from the genome of invading phages that enables CRISPR (cr)RNA-guided targeting and degradation of the phage nucleic acid upon subsequent infection. The authors describe spacer acquisition of a Type II CRISPR-Cas system when the host is challenged by phage in solid media. The structured environment of the solid media enables repeated encounters between phage and host, which increases the acquisition of multiple spacers in cases where the initial spacer was suboptimal. The latter mode of immunity leads to the emergence of greater spacer diversity in the population, increasing the likelihood of surviving subsequent challenge with escaper phage.

Essential revisions:

1) Please clarify the mentions of priming (subsection “CRISPR expansion involves priming by the first acquired spacer”, subsection “The ability of phage to escape targeting by the founding spacer determines colony heterogeneity” and the Discussion section). The conclusion of priming seems exaggerated given that the majority of spacers are spread throughout the genome and likely are not primed. For subsection “The ability of phage to escape targeting by the founding spacer determines colony heterogeneity”: The caveat provided is unclear. As the authors mention in the Discussion section, is it not expected that most spacers acquired following phage escape are through priming during colony formation? If this is the case, why is it necessary to choose an escaper phage that can no longer be targeted by Cas9? In addition, although there are two mutations present in the escaper phage (one each in the PAM and seed), the PAM mutation is 5'-NAG-3'. G to A substitutions in the PAM are known to be tolerated by S. pyogenes Cas9 (e.g. see PMID: 27041224). While the efficiency of Cas9 binding to the escaper target is likely significantly decreased, it remains possible that this crRNA can still direct Cas9 to bind the phage DNA. At this point, priming in type II-A systems is not characterized well enough to know for certain whether or not this escaper could be targeted for escaper-mediated priming.

2) Please correct the labeling of Figure 6. There appears to be a is miscommunication between the labelling of Figure 6 and the description in subsection “Host-phage co-evolution within multi-spacer colonies leads to increased resistance”; i.e. it says that F is initially resistant to phage but then has small decrease in immunity-score 1 to 0.83-referencing Figure 6A, but the figure showing those data is 6B, which is labelled as the C founder strain; similarly, also in this subsection is not describing Figure 6E.

---

## [Author Response]

Essential revisions:1) Please clarify the mentions of priming (subsection “CRISPR expansion involves priming by the first acquired spacer”, subsection “The ability of phage to escape targeting by the founding spacer determines colony heterogeneity” and the Discussion section). The conclusion of priming seems exaggerated given that the majority of spacers are spread throughout the genome and likely are not primed.

We disagree with the idea that “spacers are spread throughout the genome and likely are not primed”. Patterns of priming in type II systems had been obtained in two independent studies, one from our group examining the exact same experimental system used in the present study (PMID: 31585845) and one from the Fineran’s group analyzing several different type II systems (PMID: 30157725). In both studies priming is quantified using double-acquisition events, plotting a histogram of the distance between the first and second acquired spacer (not “throughout the genome”). While it is true that distances have a wide distribution, in all studies there are many more second spacers located within a short distance (~ 1kb) of the first spacer, than second spacers at longer distances. In both studies this have been accepted as a measure of priming in type II systems, and it is the same result we show in Figure 2C. Therefore, we believe that our conclusion, that the “result suggests that primed, or cleavage-mediated, spacer acquisition plays an important role to create multi-spacer cells and colonies”, is valid.

Having clarified this, we agree that not all sections where we referred to priming conveyed the same message. Therefore, we have modified the text as required.

Introduction (we deleted the reference to priming):

“In these colonies the new spacers are selected by the escaper phage, which lyses bacteria containing only the founder spacer and thus promotes the formation of sectored colonies.”

Subsection title:

“Priming by the first acquired spacer is likely involved in CRISPR expansion”

Discussion section:

“The high proportion of second spacers that are located within 1 kb of the founder spacer target site (Figure 2C) suggest that this is the preferred mechanism, although more detailed work will have to be carried out in the future to decisively understand how multiple spacers are acquired within one colony.”

For subsection “The ability of phage to escape targeting by the founding spacer determines colony heterogeneity”: The caveat provided is unclear. As the authors mention in the Discussion section, is it not expected that most spacers acquired following phage escape are through priming during colony formation? If this is the case, why is it necessary to choose an escaper phage that can no longer be targeted by Cas9? In addition, although there are two mutations present in the escaper phage (one each in the PAM and seed), the PAM mutation is 5'-NAG-3'. G to A substitutions in the PAM are known to be tolerated by S. pyogenes Cas9 (e.g. see PMID: 27041224). While the efficiency of Cas9 binding to the escaper target is likely significantly decreased, it remains possible that this crRNA can still direct Cas9 to bind the phage DNA. At this point, priming in type II-A systems is not characterized well enough to know for certain whether or not this escaper could be targeted for escaper-mediated priming.

We agree that our speculation about the effect of the two target mutations cannot be properly supported by currently limited knowledge about the type II priming mechanism. We also agree that the caveat is not clear. We have eliminated this passage to avoid confusion:

“We isolated a phage that can overcome targeting by spacer F0 (Figure S2), a mono-spacer colony founder, and repeated the experiment of Figure 1F infecting with a mixture of wild-type and escaper phage. We previously measured that the frequency of escape of spacer F0 is ~ 10^-8^ (Figure 3A); we added escaper phage to raise this rate to 10^-5^. As opposed to the previous results, the increase in the proportion of escaper phage generated multi-spacer colonies from this otherwise mono-spacer colony founder (Figure 3C) and contained phage that escaped the immunity provided by the F0 spacer (Figure S3B).”

2) Please correct the labeling of Figure 6. There appears to be a is miscommunication between the labelling of Figure 6 and the description in subsection “Host-phage co-evolution within multi-spacer colonies leads to increased resistance”; i.e. it says that F is initially resistant to phage but then has small decrease in immunity-score 1 to 0.83-referencing Figure 6A, but the figure showing those data is 6B, which is labelled as the C founder strain; similarly, also in this subsection is not describing Figure 6E.

Thank you for pointing this out. Figure 6A and 6B have been switched. We have corrected the figure and text.